# Beyond the Pandemic: COVID-19 Pandemic Changed the Face of Life

**DOI:** 10.3390/ijerph18115645

**Published:** 2021-05-25

**Authors:** Shaden A. M. Khalifa, Mahmoud M. Swilam, Aida A. Abd El-Wahed, Ming Du, Haged H. R. El-Seedi, Guoyin Kai, Saad H. D. Masry, Mohamed M. Abdel-Daim, Xiaobo Zou, Mohammed F. Halabi, Sultan M. Alsharif, Hesham R. El-Seedi

**Affiliations:** 1Department of Molecular Biosciences, The Wenner-Gren Institute, Stockholm University, S-106 91 Stockholm, Sweden; 2Department of Chemistry, Faculty of Science, Menoufia University, Shebin El-Kom 32512, Egypt; swilam2040@gmail.com; 3Department of Bee Research, Plant Protection Research Institute, Agricultural Research Centre, Giza 12627, Egypt; aidaabd.elwahed@arc.sci.eg; 4National Engineering Research Center of Seafood, School of Food Science and Technology, Dalian Polytechnic University, Dalian 116024, China; duming@dlpu.edu.cn; 5Faculty of Medicine, Riga Stradins University (RSU), LV-1007 Riga, Latvia; haged.hr@gmail.com; 6Laboratory of Medicinal Plant Biotechnology, College of Pharmacy, Zhejiang Chinese Medical University, Hangzhou 310053, China; kaiguoyin@zcmu.edu.cn; 7Department of Plant Protection and Biomolecular Diagnosis, Arid Lands Cultivation Research Institute (ALCRI), City of Scientific Research and Technological Applications, New Borg El-Arab City 21934, Egypt; saad.masry@adafsa.gov.ae; 8Abu Dhabi Agriculture and Food Safety Authority (ADAFSA), Al Ain 52150, United Arab Emirates; 9Pharmacology Department, Faculty of Veterinary Medicine, Suez Canal University, Ismailia 41522, Egypt; abdeldaim.m@vet.suez.edu.eg; 10School of Food and Biological Engineering, Jiangsu University, Zhenjiang 212013, China; zou_xiaobo@ujs.edu.cn; 11Al-Rayan Research and Innovation Center, Al-Rayan Colleges, Medina 42541, Saudi Arabia; m.halabi@amc.edu.sa; 12Department of Biology, College of Science, Taibah University, Al-Madinah 887, Saudi Arabia; Ssharif@taibahu.edu.sa; 13International Research Center for Food Nutrition and Safety, Jiangsu University, Zhenjiang 212013, China; 14Pharmacognosy Group, Department of Pharmaceutical Biosciences, Uppsala University, Biomedical Center, Box 574, 751 23 Uppsala, Sweden

**Keywords:** COVID-19, economy, lifestyle, health, education, environment

## Abstract

The COVID-19 pandemic is a serious challenge for societies around the globe as entire populations have fallen victim to the infectious spread and have taken up social distancing. In many countries, people have had to self-isolate and to be confined to their homes for several weeks to months to prevent the spread of the virus. Social distancing measures have had both negative and positive impacts on various aspects of economies, lifestyles, education, transportation, food supply, health, social life, and mental wellbeing. On other hands, due to reduced population movements and the decline in human activities, gas emissions decreased and the ozone layer improved; this had a positive impact on Earth’s weather and environment. Overall, the COVID-19 pandemic has negative effects on human activities and positive impacts on nature. This study discusses the impact of the COVID-19 pandemic on different life aspects including the economy, social life, health, education, and the environment.

## 1. Introduction

COVID-19 is a disease that causes the novel acute respiratory syndrome coronavirus 2 (SARS-CoV-2), which has spread throughout most countries and has caused multiple health and social problems. Disturbances in various areas of life have been consequences of the restrictions imposed by governments and health authorities, as mentioned in Table 1 [1]. For instance, health, education, community relationships, and availability of food and jobs have been impacted by the restrictions applied to limit the transmission of the disease [2].

Globally, governments and agencies (e.g., the World Health Organization (WHO)) took action to contain the spread, including minimizing travel and close contact at home, at work, and in public places [3,4]. Among the successful actions are the routines applied for healthcare and social distancing [5]. Still, these strategies have consequences, such as economic crises due to the closure of businesses and the termination of work in various industries. The quality of education has been affected by the global lockdown as governments direct educational process towards online curriculums but schools in some countries lack network platforms and services. In this vein, social media has served as the means of disseminating information used by states, organizations, and individuals. Additionally, the COVID-19 pandemic has had an impact on transport, leading to a decline in fuel consumption around the world. Reduced fuel and energy consumption has had beneficial impacts on Earth’s weather and the environment, such as a decrease in gas pollution (e.g., CO and NO_2_) and an increase in the thickness of the ozone layer [6]. 

This review highlights the impact of the COVID-19 pandemic on various sectors of life, including the economy, lifestyles, education, social networking, health, food supply, travel, the weather, and the environment. In this review, we try to resonate with the readers in regard to whether various measures and the continuous changes made due to the occurrence of the COVID-19 pandemic changed daily life for the better or worse. One of the aims is also to re-evaluate our efforts in defeating the crisis and in building more flexible societies after the pandemic. Taken together, this review illustrates how human behavior has evolved and developed in response to the spread of the virus and how humans experienced its consequences: the good, the bad, and the ugly.

**Table 1 ijerph-18-05645-t001:** Some negative and positive consequences of the COVID-19 pandemic.

Life Sectors	Negatives	Positives	References
Economy	Massive losses of money, layoffs, more tension in all sectors	Learning lessons and making plans for similar future occurrences	[7,8]
Lifestyle	Tensions; domestic violence; isolation; loneliness, especially with elder people; loss of physical activity; lack of supplies among poor people; and increased internet demands	Staying home, more sleep, less stress, more time with families, and less traffic	[9,10,11,12,13,14,15,16,17,18]
	Working from home is a double-edged sword with faithful comfort but stress in daily home routines.		[19]
Transportation	Mainly economic and financial loss	Encouraging the use of green transportation and educating about its consequences	[20,21,22,23]
	This was reflected in energy, as the quantities of fuel consumption decreased due to less flying and less use of transportation, etc. but a slower transition to green deal was also presented as economies put money into the pandemic.		[6]
Internet and social media	Misleading information	Medical use of social media, the appointment of documented pages for international health organizations, rapid transmission of medical instructions, some economies saved, and e-learning	[24,25,26,27]
Education	Higher drop-out rates of students and medical educational institutions, with the focus only on COVID-19 and fear of the future among students, and closed schools	The transition to e-learning and paying attention to infrastructure plans for pandemics in the future, especially in developing countries	[28,29,30]
Research activities	Research routines stopped at most universities and research centers	Research focus on vaccine production, high financial support for points related to COVID-19, and open access journals made free	[3,31]
	Rapid publication shared the downsides of scientific inefficiency and the advantages of quick access to solutions.		[32]
Health	COVID-19 as a new disease, psychological effects, fear of disease and infection, anxiety, obesity, eye problems due to long durations sitting in front of screens, many deaths among health workers, and extreme pressure on the global health sector	Races against time, the use of new technologies, revolution in intertwining biotechnology and medical sciences, quick vaccine production, and digitalization of health systems	[33,34,35,36]
Environmental aspects	Wildlife as a source of energy and wildlife tourism	Nature recovery, reduction in heat emissions and toxic gases (NO_2_, SO_2_, CO, CO_2_, etc.), recovery of the ozone layer, and water cleanliness (Italy and India)	[37,38,39,40,41]

## 2. Global Economic Recession

### 2.1. During the Peak Period of the COVID-19 Outbreak

The spread of the COVID-19 pandemic began at the end of 2019 in China, and the severity of the spread around the world intensified at the beginning of 2020, particularly in China, Italy, the United States, Spain, Germany, and Iran [42]. The uncontrolled pandemic prompted several governments to adopt strict regulations to prevent further infections. These regulations included quarantines and the shutdown of schools, universities, companies, and factories in a manner that has not been witnessed before [43]. Quarantine is indispensable to control further transmission of the virus, but on the other hand, it severely hampers business activities even to the point of threatening to crash the world economy [44]. The modern economy is a highly complex web of interconnections between employees, firms, suppliers, consumers, banks, and financial intermediaries. If even a few connections between any of these parts are interrupted by governmental disease-intervention policies, the outcome may provoke a cascading chain of disruptions [45].

Many studies proved that economic growth is greatly related to population health [46]. COVID-19 may cause either the death of workers or their incapacitation, and both cases affect the economy by interrupting production. Various countries previously imported numerous products from China. Since the virus outbreak, however, China’s production has considerably slowed down. Hence, the economy dropped by 0.4%, accompanied by a decrease in the global economy by 0.1% [47]. As a reaction to the growing fear, China’s central bank pumped out about USD 22 billion into the system in February 2020 to stabilize the market [48]. It will be difficult for governments to minimize the negative impact of the COVID-19 pandemic on the economy since the highest priority was to reduce death rates. However, parallel measures have to be taken to counter the inevitable economic downturn [49].

In contrast to the situation in 2008, the budgetary and banking systems grew stronger. Thus, they can react to stresses from the real economy without provoking a crisis. During the COVID-19 pandemic, the world faced a real economic shock, not only one restricted to the financial sector. The challenge is a global pandemic, not focused on low-income countries or the simultaneous reduction in demand and supply. However, some countries and companies have been affected more than others, depending on the economic structure of each region [7]. Companies undergoing issues with debt (BIS 2019) in recent years are especially vulnerable to cash flow reductions. The Flybe airline in Britain is a typical example of bankruptcy. Events such as these act as a kick-off of downward cascades. In reality, one company’s bankruptcy could put other companies at risk [45]. Below are some examples of the economic recession that affected different sectors and countries in the past few months:-One-fourth of Italy’s Gross Domestic Product (GDP) affected by emergency lockdown;-Some of the largest car manufacturers in the world suspended their production in Europe, such as Volkswagen and Ferrari;-Airbus production stopped in France and Spain;-Five million workers in China lost their jobs;-85% of Norwegian Air canceled its flights and temporarily laid off 90% of their staff (7300 employees during 16 March 2020) [50];-As a result of the lockdown period of the pandemic, some famous tourist destinations (i.e., in Paris, Rome, Venice, and Madrid) were completely deserted [50];-China’s industrial production declined by 13.5% during January and February [45].

### 2.2. After the Shock Period and Updates

After the first sudden wave, the disease spread around the world and began to change between countries. Per the latest updates of 23 April 2021, the United States harbors the largest number of injuries, followed by India; then Brazil; then France, Russia, Turkey; and the United Kingdom [51]. At equilibrium again due to the strong lessons learned during the pandemic, China was the first country to recover and then its effects spilled over to nearby countries by extension. Economic growth was observed to have increased by around 0.17% in upper-middle-income countries, i.e., India, Brazil, the Philippines, Indonesia, Malaysia, Mexico, Peru, South Africa, Thailand, and Turkey, and by 15% in high-income countries, i.e., Australia, Canada, those in the European Union, Japan, Korea, Norway, the United Kingdom, and the United States [8].

Although it is impossible to precisely determine the economic damage caused by the COVID-19 outbreak, all economists agree that it has had a severe negative effect on the global economy. Since the virus became a global pandemic, it is estimated that most economic growth will decrease by at least 2.4% of respective GDPs globally.

### 2.3. Transportation

Wuhan city, with a population of more than 11 million, is considered one of the largest cities and the hottest traffic spot in Central China. Coincidentally, the Chinese spring festival, which is associated with the largest population movement in the year, started at the same time as the COVID-19 outbreak. Five million individuals entered and/or left the city during that time. Wuhan’s transportation network was not sufficiently supplied with enough information about the new virus, thus fighting the infection and preventing its transmission were inactive. As a consequence and since the incubation period of the virus was still unknown, the virus started to spread to other Chinese cities and reached 330 cities as of 9 February 2020 [52].

The COVID-19 pandemic became a global situation within weeks. During that period, worldwide communities looked to transportation systems, public and private, as one of the main reasons for the global pandemic. As a result, transportation systems struggled to preserve their economic value due to both the preventive measures taken by governments as well as the fear of people regarding travel due to the risk of transmission/infection [22,53,54]. An inter-county ban on China was the first precaution, and hence, the challenge became a reality; this was followed by a complete suspension of planes leaving from China.

Losses to the commercial aviation sector reached nearly USD 252 billion in 2020 [21,23]. The new situation and the exacerbation of losses led to an urgent necessity to open transportation again, but at the forefront were social distancing, wearing masks, personal hygiene, and frequent hand washing, as some of the prerequisites to maintain health safety.

With regard to companies participating in the transport sector, especially aviation, the proposals began to call for return flights [55]. The aim was to accelerate the compensation for large financial losses. However, this was not initially based on a certain scientific system as a result of the confusion surrounding information. Measuring the temperature of passengers before boarding the plane, leaving middle seats empty, and using antiseptic spray and hand sanitizer in gate areas and onboard became a routine procedure used by leading airlines [55]. However, was that enough, especially considering that COVID-19 could have been transmitted from a person without any symptoms? Balancing the equation, preserving the safety of passengers, and even the areas to which passengers were transported and the continuation of navigation movement, was the critical control point. Different rules were included: reducing numbers, enforcing biosecurity, reducing the luggage to reduce the possibility of contamination, confining passengers, minimizing physical contact, limiting movement in the cabin during flight, increasing the quality of cleaning in the airplanes, simplifying the food served during flights, doubling the biosecurity check-in upon arrival at airports, as well as using the airline’s website instead of offices or corporate locations in the airports to prevent physical contact for reservations and inquiries [22,54,56]. 

Public transport simply opposes the concept of protective social distance. Frankly, some governments such as the national government of the Netherlands, have advised against exposure to public transportation except when necessary, despite prior knowledge of the economic impact and possible financial losses. In order to learn from the pandemic in transportation as well as the potential benefit of other areas, planning for future pandemics is needed. Social distancing, physical activity, and switching to green means of transport such as cycling and walking are being trialed in various cities for future application. Some cities, such as Berlin and Vienna in Europe; Philadelphia, Funko Fair, and Mexico City in North America; as well as South American cities such as Bogota, have already started to establish this idea [20,22].

Undoubtedly, other transportation proposals exist that can potentially compensate for the negative effects caused by a third wave. On the other hand, even if the damage is economic, this does not negate the damage caused by using transportation and by secondary pollution.

## 3. Lifestyle during the COVID-19 Pandemic

### 3.1. Staying at Home

The spread of COVID-19 around the world led to increased calls from many parties, including international organizations, government institutions, and individuals, to stay at home [57]. Stay-at-home orders can reduce activities associated with community spread of the virus, including population movement and close person-to-person contact outside the household. Close contact between family members, relatives, or friends is one cause of COVID-19 spread [7,8]. Stay-at-home orders might assist in limiting potential exposure to COVID-19 and garnered significant public support. However, the stay-at-home arrangement for COVID-19 patients required a well-ventilated single room and follow-up of health precautions such as consistent upkeep of hand hygiene and usage of medical masks. Additionally, it is necessary to avoid contact with other family members [18]. On the contrary, the social and psychological damages were tangible and created a feeling of isolation and loneliness [10,58]. For instance, social distancing sometimes reflected negatively on the relationships between family members. Furthermore, tensions between family members often became exacerbated because of the long unusual period of staying together, causing violence to appear (Table 1). Cases of abuse of a partner and/or children have been recorded especially in families that suffer from poverty or drug addiction. Reports on domestic violence all over the world started to increase. In China, the number of domestic violence cases tripled [11]; in France, it increased by 30% [12,59]; and in Brazil, it increased by 40–50% [60]. Many other countries have noticed an increase in domestic violence, i.e., Italy and Spain [13]. The United States also suffered from increased social violence rates during the COVID-19 pandemic, wherein cases of domestic violence accounted for a share of about 80% of the victims [15,16]. The hosting of patients in nursing homes is also difficult since COVID-19 assessments are not widely seen and not all nursing homes have doctors or nurses [61]. Moreover, a lack of access to personal protective equipment and insufficient testing of nurses in care homes poses a significant risk of transmitting the virus [62,63]. On the other hand, staying at home is accompanied by adverse consequences such as household-schooling of children, restricted outings, increases in homework or working hours in difficult situations, and limiting social contact in parallel to close control of the health risks. All of these factors can have a significant effect on day-to-day functioning and nighttime sleep. Working at home has obviated the need for a new circulation of increased internet demands, particularly for smart work, schooling, trade, social activities, and entertainment [14], but it has resulted in a major reduction in road traffic [19]. Staying at home has affected the price of food, as there were price increases in 31 European countries between January and May 2020. The most substantial food groups were beef, fish, seafood, and vegetables. The prices of bread, cereals, fruit, milk, cheese, eggs, oil, and butter have not greatly changed [4]. The food price situation in China has been more secure, although the costs of pork and cabbage have risen dramatically [64]. However, home confinement due to the COVID-19 pandemic has had a negative effect on all levels of physical exercise. Based on an international online survey (ECLB-COVID19), which opened on 1 April 2020, where 1047 replies from Asian, African, European, and others countries were received, normal sitting time increased from 5 to 8 h every day, and food intake and diet habits became more unhealthy [17]. Regular low/medium high-volume training, together with a 15–25% reduction in caloric intake, is recommended for the maintenance of neuromuscular, cardiovascular, metabolic, and endocrine health. Aerobic exercises as well as exercises of the major muscle groups are also advised [5,16]. In particular, a significant decline in social involvement, caused by forced domestic confinement, has been linked with lower levels of satisfaction. Conversely, social interaction through modern media has significantly increased in relation to the confinement period [65]. Herein, some suggestions were made to help people staying at home:-Daily food should contain fresh vegetables and protein while minimum daily consumption of calories was warranted [66,67];-Avoid frequent meals and processed foods [68];-Maintain regular bed hours [9];-Regular exercise is important to the body, and the order and duration of the physical activities should be maintained [69];-Computers, smartphones, and TV viewing should be monitored [9,17];-Plan to spend quality time with family [70,71];-Follow the WHO’s (World Health Organization) guidelines for staying at home [4].

### 3.2. Social Media

Social media has a wide-ranging definition, but the most common definition refers to social media as the platforms that connect people via the internet [72]. The most popular are communications websites such as Facebook, Instagram, Twitter, YouTube, and LinkedIn (Figure 1) [72]. Furthermore, the application of TikTok has had a large share of participation and ventilation during the period of the pandemic, as many young people and even actors have resorted to it. The latest TikTok stats show that the platform has 689 million monthly active users worldwide as of January 2021 [73]. Social media has broken into almost all areas of society, including marketing, tourism, education, and medicine [24,25,27,74,75]. Under the COVID-19 pandemic, social media has been engaged directly as the fastest method for spreading information. Both true and false information appeared online at the beginning of the pandemic. Lack of awareness about how to deal with a new virus was certainly a major problem, especially among healthcare workers. Knowledge exchange was also needed between nations, as at least 160 countries had confirmed cases as of the beginning of 2020 (March 2020). Whether it is used as entertainment or as a source of reliable or fabricated information, online communication occurs primarily through Facebook (79% of internet users), followed by Instagram (32%), and then Twitter (24%) [76,77]. The number of global social media users reached 3.6 billion in 2020 and is expected to reach 4.41 billion users in 2025 [78]. Therefore, social media could serve as a suitable method for communicating the best practices for preventing COVID-19 spread. Other statistics websites and scholarly publications are known to be slow [76]. In addition to the increase in infected cases of frontline healthcare workers, they were very preoccupied with providing care to patients. Hence, information dissemination was mainly conducted via social media, and this was performed also by international health-certified information organizations. The WHO, Centers for Disease Control and Prevention, many specialized journals, and other organizations posted information and updates online and/or on social media platforms. Facebook and Twitter directed people to confirmed medical care websites. Google Scholar highlighted the top leading journals and recommended articles surrounding COVID-19, which helped to control online traffic by directing users to trusted sources (Figure 1) [26,79,80]. Learning from the SARS pandemic and from specialists who helped manage it, social media websites displayed infographics, pop-ups, and banners informing people of protective practices. Social media started to play a virtual helping role in medical care as a diagnostic tool by leading the process of when to be tested and how to deal with the situation. Using social media along with artificial intelligence, the Alibaba group have made applications that, based on cases parameters such as self-reported health status, and history of travel and contacts, can identify COVID-19 cases [81]. Moreover, it was possible to provide psychological first aid through Chatbots, providing necessary assistance to the elderly in particular to compensate for the lack of human capabilities. Social media has made it quick and easy to create a culture of readiness for facing such anxiety [26,76].

A bad reputation for misinformation is dominant around social media. This becomes very clear when the general population addresses questions such as whether COVID-19 was created in a laboratory, what the symptoms and nature of the virus are, as well as any precautions that should be taken against it. Three studies were performed in the UK in the form of questionnaire surveys to investigate online social media use. Two of them were conducted using stratified random samples of a recruited panel (*n* = 2250 and *n* = 2254), and the third used a self-selecting sample (*n* = 949). The questions largely asked about social media usage, conspiracy beliefs, and health-protective behaviors with regard to COVID-19. All three surveys showed a negative relationship between social media and protective behaviors of COVID-19 [76,82]. Thus, social media offers many roles and uses but needs to be well controlled so that misinformation does not spread but rather serves as a means to help. 

## 4. Education and Research Activities during the COVID-19 Pandemic

### 4.1. Education

The COVID-19 pandemic brought about new strategies for dealing with education in particular. After the global spread of COVID-19, education and educational institutions have been severely affected. Global lockdown appeared to be the only way to limit the spread of infection. The first measures were taken in the Chinese city of Wuhan, which has suspended all activities, including the complete closure of all educational sectors. After successive events, the task of governments was to control the education crisis. In the beginning, it seemed that the educational semester would be postponed. In Ontario, Canada, the idea was well supported, especially when some initial indications of a school closure were quickly simulated with results indicating a decrease in the severity of the pandemic at a rate of 7.2 to 12.7% within 3–16 weeks of schools’ closure [29,83,84,85]. Despite it not being new, online courses were the only way to replace face-to-face contact. Some educational facilities already had the infrastructure needed for applying for online courses at a large scale. This can be taken as a positive point in developing countries to begin paying attention to providing these means, to consider it among essential needs, as well as to face the current crisis and to explore new strategies in the future to keep up with technological improvements (Figure 2) [28].

Medical schools were one of the sectors affected by the consequence of COVID-19. All departments in hospitals were utilized to contribute towards COVID-19 relief. This created two requests: one of them involving more doctors and even undergraduate trainee medical students because of the increasing number of infected cases that can be a learning experience in how to deal with such situations [86] and the other one being a problem that most of the undergraduate students were now focused on dealing with the COVID-19 pandemic rather than other clinical courses. A negative effect is expected for their exam scores. Students who are about to graduate but did not study some essential courses are expected to be affected in their actual work within different fields of specialization [30]. Therefore, the medical community must develop alternative plans to deal with these cases of infection. 

### 4.2. Egyptian Education System as an Example

One of the pioneering methods that were applied before the COVID-19 pandemic is providing e-books readable on tablets to each student. These technologies were tested in pre-university education in just the first two years of high school. When the COVID-19 arrived at large in Egypt, it served as an optimal time to measure the effectiveness of this technology on a large scale at all educational stages in Egypt. After school closures in Egypt, the Ministry of Education immediately took some steps to solve the problem of the pandemic by using new online platforms, i.e., Edmodo, besides developing old ones such as the Ministry’s website and the Egyptian Knowledge Bank. For the students with tablets and online systems, it was not a problem, but for others who were accustomed to the traditional education methodology, it was. This allowed students to gain new qualifications that may be useful in the future due to education moving away from fixed curricula that may be somewhat outdated compared to continuous development [87,88,89,90].

### 4.3. Research Activities

Just as life activities were influenced by COVID-19 circumstances, so was research activity. During the pandemic, most research and research-related activities have been carried out remotely/virtually. Worldwide, universities, and research centers have abruptly stopped their routines. Scientists were worried about the progress of their research projects, and in response, many universities and research institutes supported their staff working at home [31]. In parallel to this, there was an urgent need to continue research activities to find a vaccine or optimal treatment for COVID-19-related disorders, so keeping research participants and staff active was necessary. As studies on animal models have been severely hindered, timely contact between animal care staff and research workers were also arranged [3]. Aligning research activities during the COVID-19 period led to minimizing time and costs, giving ample chance to increase cooperation to reach high rates of publication and reduced viral transmissions in research participants or workers [91]. Additionally, open-access journals could be freely accessed, which allows for much progress to be made in regard to scientific trials and helps expand some research areas. During the COVID-19 pandemic, researchers followed alternative strategies including those listed below:-Preparing and writing grants, review articles, and paper submissions [3,92,93];-Administering online questionnaires [94,95];-Carrying out meta-analyses of relevant literature and research in the field [96].

As a result of these strategies, the number of publications, with a particular emphasis on pathogenic characterization and treatment of COVID-19, increased. During January and February 2020, more than 500 scientific articles were published and the number of published articles has increased every week [97]. China has the largest number of publications, while Singapore has the highest number of publications per million people [32]. 

On the other hand, some funds were directed to research activities that concern COVID-19 and its impacts (i.e., related to health, social effects, economics, and education). Government agencies such as the NIH have invested USD 1.8 billion in financing/promoting studies on COVID-19 [98], and Pennsylvania State University supported 17 proposals, with an estimated total cost of USD 1.2 million in March 2020 [3]. The treatment and prevention of COVID-19 infection have become a priority to researchers around the world. Research activities dealing with the development of PCR kits, enzyme-linked immunosorbent assay (ELISA) kits, amplification tests, serological tests, and lateral flow tests to diagnose patients suffering from COVID-19 were also planned [99]. The planned protocol involves the detection of COVID-19, and the evaluation and verification of its potential active molecules [100].

### 4.4. Research Activities and Gender Imbalance

The lockdown and restrictions in academia have caused female researchers to pay a higher price than male researchers. This was evident in the scientific performance of the female demographic, which was inversely proportional to the societies’ fight against COVID-19. In many countries, 50% of medical graduates are females and 70% work within the medical sector. However, these numbers are not reflected in the publishing rates during the COVID-19 period compared to males [101]. The rate of publication as a first author for women increased from 27% to 37% in medical journals during the period from 1994 to 2014 [102]. COVID-19 threatens this progress through inflation of the existing gender disparities. Gabster and colleagues [101] conducted a statistical analysis of 159 papers on the COVID-19 virus, and the results of the total number of females participating in the research were 9% lower than in the 2018 medical journals. Consequently, it is imperative to support women in the current pandemic until its eventual passing. The high pressure causes women to leave academic work and to look for opportunities to earn a living elsewhere or to work at home. An increase in gender disparity is detrimental and authorities should counteract it [101].

## 5. Health Sector during the COVID-19 Pandemic

### 5.1. Food Supply, Dietary Patterns, Nutrition, and Health

In the face of the COVID-19 outbreak, a set of measures have been imposed to contain the infection and to help flatten the curve. These measures include lockdowns of private and public institutions, quarantines, social distancing, and restrictions. Such measures are crucial, but they have severe effects on food availability and utilization [15,26]. The agricultural sector and food markets have encountered disruptions due to the lack of a sufficient workforce caused by restrictions to peoples’ movements and by changes in demand for food provoked by restaurant shutdowns and financial strains [103].

The health crisis struck both developed and developing countries, in which people are prone to malnourishment and starvation. In developing countries, food insecurity is mainly attributed to the rising food inflation (inability to afford adequate food supply) [104]. For instance, the lockdowns in Zimbabwe caused food prices to increase by 94.8% and food availability to decrease by 64% [35]. Nepal, which was already suffering from malnourishment and food insecurity, has yet to face the full consequences of the outbreak, which will worsen their extensive losses [105]. The State of Food Security and Nutrition in the World, in their last published edition, estimated that the COVID-19 pandemic may have added 83–132 million more people to the undernourished population of 2020 [106]. Continuous exposure to malnourishment and hunger in the long term can trigger infection development; cognitive-developmental deficits in infants; and psychological and behavioral problems such as stress, depression, and suicide. Furthermore, it can lead to chronic diseases (e.g., asthma, hypertension, diabetes mellitus, hyperlipidemia, and obesity) and can weaken the immune system [105].

In developed countries, however, it is more pertinent to trade restrictions and currency deflation [104]. Export-restrictive measures implemented by some countries have put the trade flow of staple foods such as wheat and rice at risk [103]. Vietnam, the third-largest exporter of rice, and Russia, the biggest exporter of wheat, implemented export restrictions [104]. Brazil, another major exporter of staples, reported logistical distributions that threaten the food supply chain. Argentina, the world’s largest soymeal livestock feed exporter, suffered from logistic disruptions in its supply chains. The country’s urban governments prevented soybean grain trucks from entering and exiting cities to limit virus transmission [107]. European exportations of salmon to China were halted after reports of COVID-19 transmission by salmon [108]. Serbia, Vietnam, and Kazakhstan imposed restrictions on their exports of potatoes, sugar, flour, and sunflower oil [109]. On the national level, the food supply chain was also indirectly affected by consumer behavior during the pandemic. Many of those who can afford additional food bought more than they needed, which can have disastrous consequences for communities at risk. Hoarding can lead to stock volatility, resulting in rapid rises in prices [110].

Food supply distribution influences the individual’s food-related behavior. This is relevant because nutrition is vital for health and well-being, especially if the immune system is involved. Moreover, restricted access to fresh food may also have detrimental effects on both mental and physical health [17]. Quarantine is usually accompanied by stress and anxiety. Many people cope with the stress by consuming more food and drinks to feel better. Stress-related eating is more likely to drive people to eat unhealthy foods such as snacks or chocolates and to drink soda, wine, and spirits [35,36]. Carbohydrate cravings stimulate serotonin releases, which improves one’s mood [111]. This mood effect is known to be proportional to the glycemic food index. It was pointed out that stress-driven eaters suffer from sleep disturbance, which further intensifies stress and food cravings [33]. Unhealthy dietary habits increase energy consumption and subsequent weight gain, which can increase the risk of obesity. Obesity is usually accompanied by inflammation, and it is a key factor for diabetes, heart, and lung diseases that raise COVID-19 complications [33]. The population around the world had different behaviors towards the food system during the COVID-19 pandemic. The dietary pattern of the adult population in Spain was evaluated during COVID-19 outbreak restrictions. It was reported that the Spanish population swayed its dietary pattern in a healthier direction by adopting the Mediterranean diet (MedDiet) and by decreasing consumption of junk food, sweet beverages, and pastries, among others. MedDiet involves increasing the consumption of vegetables, fruits, olive oil, and legumes, which boost the immune system. If maintained over the long run, this shift could prevent the development of chronic diseases and the complications associated with COVID-19 [112]. MedDiet was found to decrease mortality and morbidity caused by diabetes, cardiovascular disease, coronary artery disease, and cancer [36]. A systematic review found that, out of 7186 subjects assigned to the MedDiet, 5168 subjects reported significant results from the MedDiet as a potent intervention in decreasing central obesity [113]. In contrast, it was indicated that the United Arab Emirates shifted its dietary pattern of a MedDiet to a Westernized diet, which consists of non-nutritious foods with high energy, saturated fats, cholesterol, and carbohydrates and is low in polyunsaturated fats, fruits, vegetables, and fiber [114]. A similar trend was observed in Italy [115], France [116], China [117], Poland [118], Kuwait [119], and the United States [120]. The results included that the COVID-19 confinement and lockdowns increased unhealthy food intake, which caused a significant weight gain.

Furthermore, fruits and vegetables are rich in micronutrients that enhance immune function. Micronutrients include Vitamin E, Vitamin C, and β-carotene, which act as antioxidants [121]. Antioxidants have anti-inflammatory activity and boost the immune system by increasing the production of T-cell subunits and interleukin-2, by activating natural killer cells, and by improving the response to lymphocytes to mitogen [33]. Additionally, Vitamin D has a protective role in the respiratory tract of three modes of action: (1) maintaining tight junctions that prevent the infiltration of immune system cells into the lungs or other organs; (2) decimation of enveloped viruses by the activation of antimicrobial peptides defensins and cathelicidin; and (3) immunomodulatory activity by decreasing pro-inflammatory cytokines production that causes inflammation that damages the lungs’ lining, leading to pneumonia [41,42,122,123].

Thus, both individuals and governments share the responsibility to shift to a preventative paradigm during the pandemic or at least to alleviate the impact of clinical symptoms in people who were already infected. Governments and donors must respond quickly as soon as possible to build a food system [124].

### 5.2. Health and Psychological Effects on Students during the School Closures

Psychological effects of the pandemic and quarantine measures are evident among all groups of people, including fear of disease and infection, as well as its consequences. Among students, fear of the future, especially concerning educational attainment and qualification for jobs, has been abundant. In China, 24.9% of university students in a sample of 7143 suffered from anxiety due to the COVID-19 outbreak, where 0.9% experienced severe anxiety and 21.3% experienced mild anxiety. The effects were more pronounced among students that were more isolated due to social distancing [125]. For children, especially those with mental problems, the daily school routine is a psychological comfort. Therefore, when schools were closed, mental health symptoms were expected to increase, and the problem was that there was no alternative [126]. If a possible solution is for children to play online games, this leads to sitting for long periods in front of a computer or mobile phone. Here, obesity comes to the forefront of problems related to inactivity and school closure in addition to unhealthy habits among children. Under these circumstances, support is needed for systematic home-based programs supervised by schools for children’s physical exercise [127].

### 5.3. Vaccine Production

Combatting the COVID-19 pandemic is a race against time, testing the world’s ability to act quickly as the virus mutates. Subsequently, a variety of efforts have been made to create vaccines against COVID-19. Vaccine production takes place at various stages, including preclinical and clinical stages, which is a three-phase process. According to the WHO, the vaccine must be highly efficient, healthy, and appropriate for all ages and backgrounds [128]. Researchers are investigating various formulations of medications for treating COVID-19 patients, but all of the formulas are still under examination. In addition to numerous marketed antiviral drugs, there are also small molecule compounds in research that have demonstrated significant inhibitory activity on several main proteins of related coronaviruses, such as SARS-CoV and Middle East respiratory syndrome coronavirus (MERS-CoV) [97].

The applications of nanotechnology will provide a major contribution to the war against COVID-19. Nanomaterial have been used for the development of point-of-care diagnostics, therapeutic carriers, and vaccine development [129]. Biosensors could detect viruses early, and nano-sized vaccines could be powerful agents to prevent viral infections [130]. Nanomaterial can interact with the whole viral particle or with the surface proteins and other structural components, leading to virus inactivation. However, regulatory issues, large-scale production, and deployment to the public are still challenges for vaccine development [131].

After nearly one year, Pfizer in cooperation with BioNTech announced (on 9 November 2020) its readiness to offer a vaccine against COVID-19 with some realistic and trusted documentation (Figure 3) [34]. The results were considered extremely rapid in relation to common vaccine production processes. Such processes normally require many years of high-quality research and procedures in follow-up, bearing in mind the viral mRNA and the potential for mutation [132]. The acceleration, in this case, was due to huge financial support, which was offered by governments and the organization to develop a new vaccine. When COVID-19 first appeared in December 2019 and January 2020, the genome sequence was unknown, yet within weeks, the identification of its structure of proteins was nearly complete. Clinical trials were the next step, and the kit process was performed within months.

Besides the Pfizer (New York, NY, United States) and BioNTech (Mainz, Germany) vaccine (BNT162b2), another three vaccines have been in development. The first was a DNA (human adenovirus vector)-type vaccine developed by Gamaleya Research Institute of Epidemiology and Microbiology (Moscow, Russia) and named Sputnik V. The others were an mRNA-type vaccine developed by Moderna (Cambridge, MA, USA; Visp, Switzerland (called mRNA-1273) and a DNA (chimpanzee adenovirus vector)-type vaccine developed by the University of Oxford and AstraZeneca (Oxford/Cambridge, England) given the name (AZD1222, also known as ChAdOx1 nCoV-19 or Covishield). In mid-November 2020, the efficiencies announced according to the developers and their data were 95% for both BNT162b2 and mRNA-1273, 92% for Sputnik V, and 62–90% for AZD1222. All of them reached Phase III clinical trials, and BNT162b2 was introduced as a vaccine for emergency use, seeking approval of the FAD [133].

At the beginning of 2021, the questions and discussions shifted from the possibility of vaccine occurrences to the extent of its effectiveness. The studies were carried out to identify the extent of the vaccines’ impact and the conditions that must be met to achieve the maximum possible effectiveness [134]. Hence, imperative action to continue this rapid approach is warranted in order to maintain the pace of new developments, whether it is new strains of the virus or the unknown, and to confirm the vaccine outcomes scientifically.

## 6. Global Warming and Reports on Weather and the Environment during the COVID-19 Pandemic

Despite the epidemic’s impact on humans, it has been somewhat beneficial to nature during the pause from human activity. Are human beings harmful to Mother Nature? Are their actions the reason for pollution in the air and in the Earth’s atmosphere? These questions were almost answered by the worldwide large-scale experiment-like quarantine while nature seemingly recovered. Modern technologies and satellites coupled with ground-based mapping helped researchers make fast predictions and conduct real-time screening of the change in the atmosphere [38,135,136].

Changes started to be noticed in the calculations after a small period post-implementation of COVID-19 precautions as a result of the reduced use of coal-fired power stations and less oil burned for transportation, and a 20% drop in the scale of greenhouse gas emissions was observed by March 2020 in China. Chinese cities reported an approximately 40% drop in comparison to the same period in 2019 [94,95]. Similar observations were made in Japan, India, and Europe. For instance, nitrogen dioxide (NO_2_) emission rates in Europe were 50% lower, according to the European Environment Agency 2020, and averaged between 20–30% in the United States. In February and March of 2020, the levels of particulate matter (PM_10_ and PM_2.5_) such as CO and SO_2_ decreased in the Latin American atmosphere, and the ozone layer increased in thickness [38].

As a result of a reduction in the use of fuels in factories and transportation, the stress on nature was relieved, and for a short period during the pandemic, the environment started to recover. For example, positive changes were seen in Earth’s water. A major effect of social distancing and quarantine was the reduction of tourism, which resulted in the notably clean appearance of beaches worldwide. Many cities all over the world, including Barcelona, Spain, Acapulco, Mexico, Salinas, and Ecuador became cleaner and had clear water. In Italy, fish could be seen again in Venice’s canals, and its water became cleaner than in the past. In India, the quality of water of the Ganga River was reported to be cleaner by 40–50% during the quarantine. Organic and inorganic human waste was reduced in all manners including personal usages and factory waste after the lockdown of industrial sectors. For instance, the total dissolved solids decreased by almost half [39,40,41]. Additionally, to ensure further precautionary practice, states ordered more disinfection routines considering the virus transmission via water. More chlorine, which resembles virus-killing reagent, makes wastewater cleaner, including factory wastewater [40]. In the United States, a third of Americans drink groundwater as their primary source of hydration and most neighborhoods are located near factories, which may increase the risk of wastewater leaking into the groundwater [137]. In this context, the pandemic has helped in a certain manner to save nature or at least to make states once again rethink issues surrounding pollution and how to solve them.

From another point of view that encourages more unconventional green solutions, some harmful habits, direct or indirect, with the considerations towards their necessity should be highlighted. Since people were quarantined and must spend most of their time at home with a semi- or complete stop in the energy production sector, poor people must use more firewood as a cheap and available source of energy. The most available sources were forests. As a result of this, not only air pollution but also deforestation is becoming issues. A 50% increase in deforested areas according to the Brazilian National Institute for Space Research (INPE) was already noticed in the first quarter of 2020 compared to the previous year [38].

## 7. Conclusions

COVID-19 poses a great threat to global health. After the aggravation of the pandemic, it is no longer just a transient health condition but rather has led to global quarantines. Countries have begun a battle of endurance in regard to confronting the disease spread. Certainly, some societies have had recent learning opportunities from near-term experiences of viruses such as SARS and MERS. However, the speedy implementation of preventive measures and public awareness that was raised through social media contributed to saving lives. Contributions to scientific and educational fields came from professionals working at home, who appreciated the opportunity to do so.

Sometimes, misfortunes come with benefits. During the crisis, the greatest benefit to the land and to nature resulted from human quarantine. The global climate recorded high cure rates and gave humans a definitive guide about how to solve some of nature’s problems that had previously been thought to be impossible. Response from nature served to alert humans and provided insights into human selfishness and negligence of nature.

Recently, some communities have begun to relax their precautionary measures again when confronting the disease. Although social distance and healthcare strategies are the main methods used to avoid disease spread, communities have noticed that these strategies have consequences for mental health; for children, and younger and elder individuals; and for the poor or those with limited income [138,139]. Negative effects extend also to food availability and the economy. Patience and rationality are very much required when making decisions during the coming period or the post-pandemic period and when returning to normal life in order to avoid experiencing a relapse in the spread of the disease. It is necessary to think outside the box and not only to identify the negative points but also to learn from them. Ideas or, more precisely, evolutionary leaps that create a better society with greater intellect and awareness of immunity always sprout during crises and even wars.

All of this prompts us to focus on what is referred to as the “green deal”, i.e., a transition to renewable sources of green energy, green economical agendas, and respect for nature and its valuable role. Politicians, world leaders, scientists, ecologists, economists, and epidemiologists should invest their time and efforts towards such goals. For instance, the United Nations Climate Change Conference should prioritize and unify efforts aimed at changing attitudes towards the environment. The COVID-19 pandemic serves as a reminder [140,141].

## Figures and Tables

**Figure 1 ijerph-18-05645-f001:**
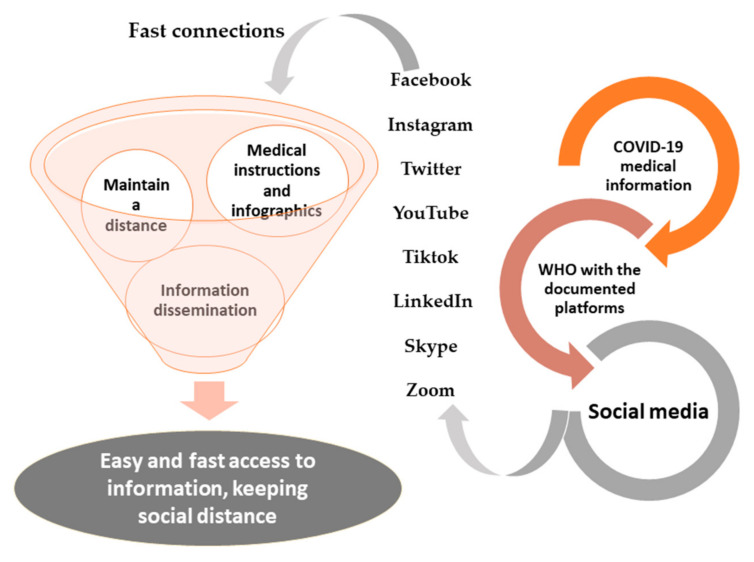
Social media participation and the continuation of its role during the pandemic, especially using documented pages of international and local organizations as well as press institutions on the Facebook and Twitter platforms to update information about COVID-19 around the clock for users.

**Figure 2 ijerph-18-05645-f002:**
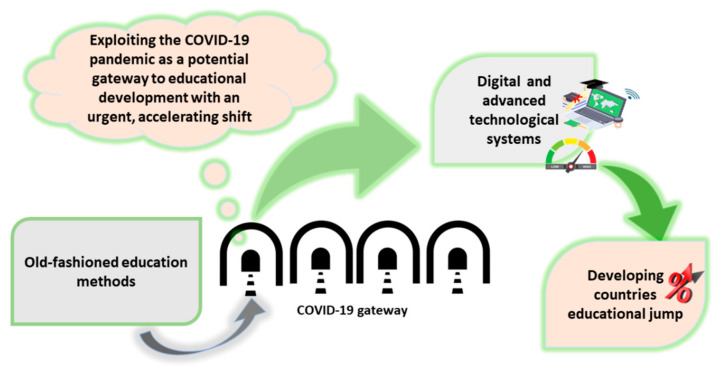
Exploitation of the COVID-19 Pandemic is one of the gateways in forcing the technological transfer and development in education as a basic need in developing countries.

**Figure 3 ijerph-18-05645-f003:**
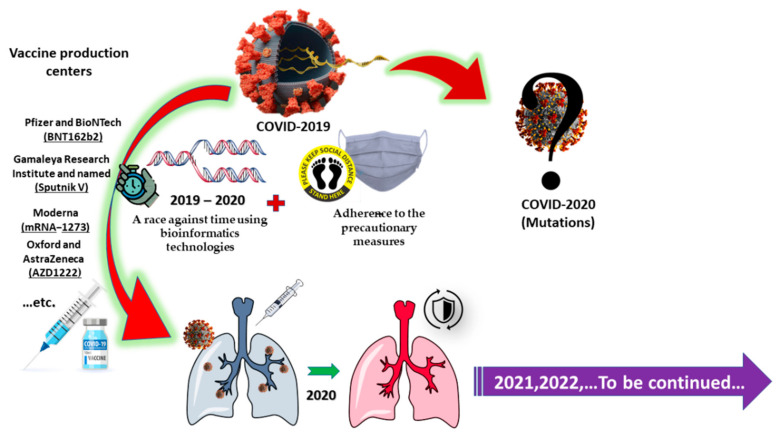
Impact of the COVID-19 pandemic on the vaccine production process and new challenges.

## Data Availability

No new data were created or analyzed in this study. Data sharing is not applicable to this article.

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
