# Peer review of "Beyond the Pandemic: COVID-19 Pandemic Changed the Face of Life"

_ijerph, 2021, doi:10.3390/ijerph18115645_

Round 1
Reviewer 1 Report
Dear authors,
Thank you for an interesting article. Please, consider the following points to improve it:
1) Please, provide references for numbers you declare in this sentence:
"In China the domestic violence increased to 92 triple the original numbers, in France it increased by 30%, in Brazil, it jumped by a 93 40-50% increase."
and also to support this statement:
"Carbohydrate 232 craving stimulates serotonin release, which improves one’s mood."
Small hints:
1) Capital letters should be used after a full stop:
"health, social life, and mental wellbeing. on other hands, due to reduced population movements 36·
2) Please, check syntaxis - why do you use a question at the end of this sentence?:
"had a positive impact on the Earth’s weather and environment? So COVID-19 pandemic has nega..."
Please, check everywhre - incongruent syntaxis errors are met in different places.
3) "CO" should be in in normal letter size:
"(e.g., CO, NO2) and an increase in the thickness of the ozone layer [6]. 63"
Please, check other innacurracies on syntaxis and spacings, as here, for example (also United States should start with capital letters:
"[50], Poland [51], Kuwait [52], and the united states[53]. The results included that the 255"
4) Word use: maybe "sistematic" reviews better?:
"- Screening critical literature or conducting systemic reviews [88]. 407"
Author Response
Comments and Suggestions for Authors
Dear authors,
Thank you for an interesting article. Please, consider the following points to improve it:
We would like to thank the reviewer for the nice words and encouragement.
1) Please, provide references for numbers you declare in this sentence:
- "In China the domestic violence increased to 92 triple the original numbers, in France it increased by 30%, in Brazil, it jumped by a 93 40-50% increase."
and also to support this statement:
- "Carbohydrate 232 craving stimulates serotonin release, which improves one’s mood."
Response: Done
"In China the domestic violence increased to triple the original numbers [12], in France it increased by 30% [13,14], in Brazil, it jumped by a 40-50% increase [15]. Many other countries have noticed an increase in domestic violence i.e. Italy, Spain [16]. USA also suffered from the increased social violence rates during the COVID-19 pandemic, wherein cases of domestic violence accounted for a share of about 80% of the victims [17,18]. "
Reference
- Allen-Ebrahimian, B. China’s domestic violence epidemic. Axios. https://www.axios.com/china-domestic-violence-coronavirus-quarantine-7b00c3ba-35bc-4d16-afdd-b76ecfb28882.html 2020.
- Guenfoud, I. French women use code words at pharmacies to escape domestic violence during coronavirus. https://abcnews.go.com/International/french-women-code-words-pharmacies-escape-domestic-violence/story?id=69954238 2020.
- EuroNews Domestic violence cases jump 30% during lockdown in France. https://www.euronews.com/2020/03/28/domestic-violence-cases-jump-30-during-lockdown-in-france/ 2020.
- Graham-Harrison, E., Giuffrida, A., Smith, H., & Ford, L. Lockdowns around the world bring rise in domestic violence. https//www.theguardian.com/ Soc. 2020.
Piquero, A.R.; Jennings, W.G.; Jemison, E.; Kaukinen, C.; Knaul, F.M. Domestic violence during the COVID-19 pandemic - Evidence from a systematic review and meta-analysis. J. Crim. Justice 2021, 74, 101806, doi:10.1016/j.jcrimjus.2021.101806.
- Usher, K.; Bhullar, N.; Durkin, J.; Gyamfi, N.; Jackson, D. Family violence and COVID-19: Increased vulnerability and reduced options for support. Int. J. Ment. Health 2020, 29, 549–552, doi:10.1111/inm.12735.
- Campbell, A.M. An increasing risk of family violence during the Covid-19 pandemic : Strengthening community collaborations to save lives. Forensic Sci. Int. Reports 2020, 2, 100089–100091, doi:10.1016/j.fsir.2020.100089.
-“Carbohydrate craving to stimulates serotonin releases, which improves one’s mood [44].”
References:
Strasser, B.; Gostner, J.M.; Fuchs, D. Mood, food, and cognition: Role of tryptophan and serotonin. Curr. Opin. Clin. Nutr. Metab. Care 2016, 19, 55–61, doi:10.1097/MCO.0000000000000237.
Small hints:
1) Capital letters should be used after a full stop:
"health, social life, and mental wellbeing. on other hands, due to reduced population movements 36·
Response: Changed
2) Please, check syntaxis - why do you use a question at the end of this sentence?
"had a positive impact on the Earth’s weather and environment? So COVID-19 pandemic has nega..."
Please, check everywhere - incongruent syntaxis errors are met in different places.
Response: Changed
"which had a positive impact on the Earth’s weather and environment? . So COVID-19 pandemic has negative effects on human activities and positive impact on the nature".
3) - "CO" should be in in normal letter size:
"(e.g., CO, NO2) and an increase in the thickness of the ozone layer [6]. 63"
Response: Adjusted
- Please, check other innacurracies on syntaxis and spacings, as here, for example (also United States should start with capital letters: "[50], Poland [51], Kuwait [52], and the united states[53]. The results included that the 255"
Response: Done
"A similar trend was observed in Italy [54], France [55], China [56], Poland [57], Kuwait [58], and the United States [59]."
4) Word use: maybe "sistematic" reviews better?
"- Screening critical literature or conducting systemic reviews [88]. 407"
Response: Changed
"Screening critical literature or conducting systematic reviews [95]."
Reviewer 2 Report
The authors review various sectors on which the pandemic left a dent. However, the manuscript needs
a) updating to April 2021 - many claims are imbalanced or outdated
b) restructuring. It is a chaotic read with partial repetitions, as if each paragraph got written by different authors (which is presumably true as the quality of the English also varies across sections).
c) instead of this not very informative figures the review should list a table of positive and negative effects per category, e.g.
Personal health: positive - more sleep, less stress, negative - anxiety etc
health system: positive - digitalization, ..., negative - many dead health workers due to COVID-19
research: positive - vaccine development, transition to open science, negative - career delays (PhD students without funding, post-docs, closed labs, gender gap increasing)
energy: positive - less flying, negative - slower transition to green deal as economies put money into the pandemic ...
education: positive - e-learning, negative - higher drop-out rates of students
these are just examples, there is far more in each sector, and the authors do not balance the consequences well. It is often just a listing, and occasionally the complexity is sketched (i.e. economic issues affecting health)
d) the abstract and conclusion differ from what is reviewed.
More details provided below.
The virus is SARS-Cov-2, the disease is COVID-19. Please correct your very first sentence accordingly, i.e. COVID-19 is caused by SARS-CoV-2, it is NOT as you write a coronavirus.
I would not say that the disturbances are spontaneous - these are consequences of the restrictions imposed by governments and health authorities
Citation: I am aware that there is a flood of papers on COVID-19 and nobody can read all of it, but certain arguments are without any citations, whereas others, more minor arguments receive citations, e.g. line 92-94 has no support of the claims that violence in China, France, Brasil increased, but the claim for the US has two citations.
there are lots of English language issues, I highly recommend to ask a colleague who is native English speaker to proof-read the manuscript.
The use age however, on the contrary etc is often inappropriate. The article "the" is sometimes not used when it should, and in other places is used when it shouldn't (e.g. line 162: Middle seat kept empty - should be: or "Keeping empty middle seats, ..."
or
Netherlands national government - should be the national government of the Netherlands
line 63: CO2 has odd font size and missing 2
line 69: "to build a more flexible after the pandemic societies" - maybe you want to say: to build a more flexible society after the pandemic?
line 148: "transportation systems were struggle in preserve their economic value" - I guess you want to say: transportation systems struggled to preserve their economic value. Though this still sounds odd, so better:
...transportation systems struggled to maintain their economic relevance and value...
line 179: future thinking is needed - that is an odd formulation, either you mean more research is needed, or smth along the lines of: planning for future pandemics is needed ...
line 181: "and walking are idea starting applying" - I guess you want to say smth like: ... are tried out in many cities.
not sure what in line 183 "this idea" refers to what? Walking, cycling, social distancing? Grammatically not clear. I guess you mean smth along the lines of: These changes in transportation have been tested in major cities like ...
but then you could rewrite the previous and this sentence in a better manner.
line 184 is also grammatically hard to parse. I have no idea what you mean by "proposals are still" - are they in the making? Sorry, but the entire sentence is not good English and I can't figure out what you mean with proposals are still. My best guess is: other transportation proposals exists which potentially can compensate for negative effects caused by a third wave ... (if proposals refers to means of transportation, which is not clear)
line 255ff "The results included that the COVID-19 confinement and lockdowns increased unhealthy food intake which caused a significant weight gain. " you cite Spain where a more healthy diet was found, so that contradicts your statement. Please be more balanced. It is - as with everything - not so easy / black and white. You have countries where restrictions / life changes during the pandemic are less severe, and other countries were it is more severe. It also differs by income. The rich stay healthy and rich, the poor are more likely to get infected, get unemployed etc. So countries with a low Gini index may have less severe consequences on an individual basis than countries with a high Gini index (incl that the pandemic increases the distance between poor and rich)
"Thus, it is the individual’s responsibility to shift to a preventative paradigm during the pandemic or at least alleviate the impact of clinical symptoms in the case of already being infected. This could be achieved through efforts to choose a healthier lifestyle, including an adequate diet"
No, this is not true. If you don't have access (think of all the starving children in Africa, Asia) you can't prevent getting ill. Here you mix up two very important concepts: what is the responsibility of the individual - and what is the responsibility of the state / society.
If there is a choice (what food to eat, how to commute, where to live) then you can say that is in the hands of the individual. But if there is no choice, it is not the individual's fault but an issue of the system.
If a country has no resources to buy vaccines, it can't vaccinate its citizens. It is not the responsibility of the individual to get vaccinate if there is no access to vaccines. Similarly, if you have no bike, boat etc, you can't say green transport is within the individual. Or take diet: if healthy food is expensive, but junk food isn't, claiming a poor person is responsible for their obesity is not correct. It is never that simple as you state.
Line 275: The most popular are communications websites such as Facebook, Instagram, Twitter, YouTube, and LinkedIn - that is quite western society centered. I would expect Alibaba to have a large if not larger community than LinkedIn. And what about TikTok?
line 283ff: Before the pandemic, 2016 statistics showed that most adults accessed information primarily through Facebook (79% of internet users), followed by Instagram (32%) and then Twitter (24%) [43, 63].
Here you have to distinguish information from entertainment. In a range of countries the pandemic caused many / majority to watch and consume trustworthy information sources like national broadcasting institutions. Part of those news might appear (as illegal copy in a strict sense, see Australia vs Facebook on news reporting) on social media.
Please be more balanced. Just using Facebook and Co, does not mean people trust those information or deem this relevant information (for example about taxes, food safety etc)
line 305 psychological first aid through Chabot - do you mean through chatbots? Never heard the term Chabot, please explain what this is. Thanks
line 308 A leader in e-learning during the pandemic is the Zoom meetings app, which allows teachers and students to simulate a live course [70]
Are you sponsored by Zoom? Admittedly, Zoom benefited from the pandemic, but there are other tools (MS teams, CANVAS) that offer e-learning, and video-chat communication.
More crucial though, you jump from social media used for providing information to citizens to e-learning. This comes very sudden. I would expect a separate paragraph on education during the pandemic (and used technology)
Again, not all countries have the resources (read: money) to use zoom.
line 367: "graduating without studying some basic courses will affect during the actual work" - I think you mean: ... courses will have an affect on their actual work ...
line 377: e-books in the form of a tablet or iPad delivered to each student - English is odd, an e-book does not have the form of a tablet or iPad - these are just means to read an e-book. Maybe say smth like: ... e-books readable on tablets (it is not needed to state iPad as an iPad is a tablet)
line 443: The WHO Director recently announced that WHO will perform clinical trial ... this is old news, the solidarity trial started in April 2020, we now have April 2021
what you also miss is the gender imbalance. The lockdown and restrictions in academia has caused female researchers to pay a higher price (home schooling) than male researchers. The long-term consequences are fewer females in top positions, which will affect recruiting, the academic culture in general, and as you know: a country with more gender balance / equity is economically more successful (see e.g. Scandinavian countries, New Zealand etc).
line 474: The acceleration in this case was due to the advent of genomics technologies and computational approaches.
This is not per se wrong, but the acceleration was mostly due to governments (EU) putting lots of money into companies developing vaccines and streamlining the bureaucracy / administration / ethical approvals that comes with vaccine development (phase I, phase II and phase III permissions)
line 482: however, as of August 11, 2020, it had not entered Phase III clinical trials - we have now April 2021, please update your manuscript
line 508: of a large drop that occurred in the atmosphere change [107–109].
either delete change or say that the change in the atmosphere was a large drop
line 568: COVID-19 has spread worldwide since the turn of the year, taking a large number of victims, particularly in China, Italy, the U.S., Spain, Germany, and Iran [117].
This seems to be a very outdated references. Most victims are in Brazil and India. Within Europe it is neither in absolute not in relative numbers Germany (first wave it was Belgium and UK, now it is France, Poland, Estland). You can use the John Hopkins dashboard: https://coronavirus.jhu.edu/ or ourworldindata.org
line 603ff: the examples you give should be updated.
1. most of these companies received state loans
2. some of these companies are back to (nearly) normal, e.g. car manufactures
entire paragraph needs updating. China is back on track, and the gross domestic product is predicted to be positive for most countries
minor typos
line 164: [30].. - remove one "."
line 169: s, , e - remove space and ,
line 180 andswitching - space missing
line 255 united states - United States
line 373: getaways - gateways
line 375: The Egyptian educational as example - presumably you want to say; the Egyptian education system as an example
line 380: on large scale with all educational stages - ... at all ...
line 385: made students gains - gain
line 593 pandemic,, th - remove ","
line 596: "have been effected than others" - more is missing, i.e. effected more than others
line 599: Brittan - do you mean Britain? UK?
line 619: 2.4%t of the GDP - remove t
there are more English issues, this paper really needs proof-reading. But since you should also restructure, update and thereby rewrite many sections, I won't list all odd sentences.
Author Response
The authors review various sectors on which the pandemic left a dent. However, the manuscript needs …..what? I do not understand this part?
Response:
- updating to April 2021 - many claims are imbalanced or outdated
Response: The references are updated throughout the manuscript (highlighted in the revised version with yellow color).
- Hamidi, S., Zandiatashbar, A., 2021. Compact development and adherence to stay-at-home order during the COVID-19 pandemic: A longitudinal investigation in the United States. Landsc. Urban Plan. 205, 103952–103960. https://doi.org/10.1016/j.landurbplan.2020.103952
- Albaqawi, H.M., Pasay-an, E., Mostoles, R., Villareal, S., 2021. Risk assessment and management among frontline nurses in the context of the COVID-19 virus in the northern region of the Kingdom of Saudi Arabia. Appl. Nurs. Res. 58, 151410–151415. https://doi.org/10.1016/j.apnr.2021.151410.
- de Faria Coelho-Ravagnani, C., Corgosinho, F.C., Sanches, F.L.F.Z., Prado, C.M.M., Laviano, A., Mota, J.F., 2021. Dietary recommendations during the COVID-19 pandemic. Nutr. Rev. 79, 382–393. https://doi.org/10.1093/nutrit/nuaa067
- Khoramipour, K., Basereh, A., Hekmatikar, A.A., Castell, L., Ruhee, R.T., Suzuki, K., 2021. Physical activity and nutrition guidelines to help with the fight against COVID-19. J. Sports Sci. 39, 101–107. https://doi.org/10.1080/02640414.2020.1807089
- Kumari, A., Ranjan, P., Sharma, K.A., Sahu, A., Bharti, J., Zangmo, R., Bhatla, N., 2021. Impact of COVID-19 on psychosocial functioning of peripartum women: A qualitative study comprising focus group discussions and in-depth interviews. Int. J. Gynecol. Obstet. 152, 321–327. https://doi.org/10.1002/ijgo.13524
- b) restructuring. It is a chaotic read with partial repetitions, as if each paragraph got written by different authors (which is presumably true as the quality of the English also varies across sections).
Response: Authors read the manuscript thoroughly and amend it accordingly. The corrections and modifications are highlighted in yellow.
- c) instead of this not very informative figures the review should list a table of positive and negative effects per category, e.g.
Personal health: positive - more sleep, less stress, negative - anxiety etc
health system: positive - digitalization, ..., negative - many dead health workers due to COVID-19
research: positive - vaccine development, transition to open science, negative - career delays (PhD students without funding, post-docs, closed labs, gender gap increasing)
energy: positive - less flying, negative - slower transition to green deal as economies put money into the pandemic ...
education: positive - e-learning, negative - higher drop-out rates of students
these are just examples, there is far more in each sector, and the authors do not balance the consequences well. It is often just a listing, and occasionally the complexity is sketched (i.e. economic issues affecting health)
Response: We would like to thank the referee once again for this comment; Fig. 1 is modified, Fig. 2 is deleted and Table 1 was added to the manuscript
- d) the abstract and conclusion differ from what is reviewed.
Response: Changed and modified
More details provided below.
- The virus is SARS-Cov-2, the disease is COVID-19. Please correct your very first sentence accordingly, i.e. COVID-19 is caused by SARS-CoV-2, it is NOT as you write a coronavirus.
Response: Done
"COVID-19 is disease that causes the novel acute respiratory syn-drome coronavirus 2 (SARS-CoV-2), which has spread through most countries and caused multiple health and social problems.
- " I would not say that the disturbances are spontaneous - these are consequences of the restrictions imposed by governments and health authorities
Response: Done
"Disturbances in the various areas of life have been consequences of the restrictions imposed by governments and health authorities as mentioned in Table 1"
- Citation: I am aware that there is a flood of papers on COVID-19 and nobody can read all of it, but certain arguments are without any citations, whereas others, more minor arguments receive citations, e.g. line 92-94 has no support of the claims that violence in China, France, Brazil increased, but the claim for the US has two citations.
Response: The citation added
"In China the domestic violence increased to triple the original numbers [12], in France it increased by 30% [13,14], in Brazil, it jumped by a 40-50% increase [15]. Many other countries have noticed an increase in domestic violence i.e. Italy, Spain [16]. USA also suffered from the increased social violence rates during the COVID-19 pandemic, wherein cases of domestic violence accounted for a share of about 80% of the victims [17,18]. "
References
- Allen-Ebrahimian, B. China’s domestic violence epidemic. Axios. https://www.axios.com/china-domestic-violence-coronavirus-quarantine-7b00c3ba-35bc-4d16-afdd-b76ecfb28882.html 2020.
- Guenfoud, I. French women use code words at pharmacies to escape domestic violence during coronavirus. https://abcnews.go.com/International/french-women-code-words-pharmacies-escape-domestic-violence/story?id=69954238 2020.
- EuroNews Domestic violence cases jump 30% during lockdown in France. https://www.euronews.com/2020/03/28/domestic-violence-cases-jump-30-during-lockdown-in-france/ 2020.
- Graham-Harrison, E., Giuffrida, A., Smith, H., & Ford, L. Lockdowns around the world bring rise in domestic violence. https//www.theguardian.com/ Soc. 2020.
Piquero, A.R.; Jennings, W.G.; Jemison, E.; Kaukinen, C.; Knaul, F.M. Domestic violence during the COVID-19 pandemic - Evidence from a systematic review and meta-analysis. J. Crim. Justice 2021, 74, 101806, doi:10.1016/j.jcrimjus.2021.101806.
- Usher, K.; Bhullar, N.; Durkin, J.; Gyamfi, N.; Jackson, D. Family violence and COVID-19: Increased vulnerability and reduced options for support. Int. J. Ment. Health 2020, 29, 549–552, doi:10.1111/inm.12735.
- Campbell, A.M. An increasing risk of family violence during the Covid-19 pandemic : Strengthening community collaborations to save lives. Forensic Sci. Int. Reports 2020, 2, 100089–100091, doi:10.1016/j.fsir.2020.100089.
- there are lots of English language issues, I highly recommend to ask a colleague who is native English speaker to proof-read the manuscript.
Response: The authors read the manuscript thoroughly and amend it accordingly. The manuscript was subjected for extensive English editing.
- The use age however, on the contrary etc is often inappropriate. The article "the" is sometimes not used when it should, and in other places is used when it shouldn't (e.g. line 162: Middle seat kept empty - should be: or "Keeping empty middle seats, ..."
or
Netherlands national government - should be the national government of the Netherlands
Response: Changed to " Frankly, some governments such as the national government of the Netherland"
- line 63: CO2 has odd font size and missing 2
Response: Adjusted
- line 69: "to build a more flexible after the pandemic societies" - maybe you want to say: to build a more flexible society after the pandemic?
Response: Corrected
"to build a more flexible societies after the pandemic."
- line 148: "transportation systems were struggle in preserve their economic value" - I guess you want to say: transportation systems struggled to preserve their economic value. Though this still sounds odd, so better:
...transportation systems struggled to maintain their economic relevance and value...
Response: Changed to “transportation systems struggled to preserve their economic value”
- line 179: future thinking is needed - that is an odd formulation, either you mean more research is needed, or smth along the lines of: planning for future pandemics is needed ...
Response: changed to “planning for future pandemics is needed”
- line 181: "and walking are idea starting applying" - I guess you want to say smth like: ... are tried out in many cities.
not sure what in line 183 "this idea" refers to what? Walking, cycling, social distancing? Grammatically not clear. I guess you mean smth along the lines of: These changes in transportation have been tested in major cities like ...
but then you could rewrite the previous and this sentence in a better manner.
Response: Changed to "Social distance, physical activity, and switching to green means of transport such as cycling and walking are tried out in many cities."
- line 184 is also grammatically hard to parse. I have no idea what you mean by "proposals are still" - are they in the making? Sorry, but the entire sentence is not good English and I can't figure out what you mean with proposals are still. My best guess is: other transportation proposals exists which potentially can compensate for negative effects caused by a third wave ... (if proposals refers to means of transportation, which is not clear)
Response: changed to “other transportation proposals exist which potentially can compensate for negative effects caused by a third wave”
- line 255ff "The results included that the COVID-19 confinement and lockdowns increased unhealthy food intake which caused a significant weight gain. " you cite Spain where a more healthy diet was found, so that contradicts your statement. Please be more balanced. It is - as with everything - not so easy / black and white. You have countries where restrictions / life changes during the pandemic are less severe, and other countries were it is more severe. It also differs by income. The rich stay healthy and rich, the poor are more likely to get infected, get unemployed etc. So countries with a low Gini index may have less severe consequences on an individual basis than countries with a high Gini index (incl that the pandemic increases the distance between poor and rich)."Thus, it is the individual’s responsibility to shift to a preventative paradigm during the pandemic or at least alleviate the impact of clinical symptoms in the case of already being infected. This could be achieved through efforts to choose a healthier lifestyle, including an adequate diet"
No, this is not true. If you don't have access (think of all the starving children in Africa, Asia) you can't prevent getting ill. Here you mix up two very important concepts: what is the responsibility of the individual - and what is the responsibility of the state / society.
If there is a choice (what food to eat, how to commute, where to live) then you can say that is in the hands of the individual. But if there is no choice, it is not the individual's fault but an issue of the system.
If a country has no resources to buy vaccines, it can't vaccinate its citizens. It is not the responsibility of the individual to get vaccinate if there is no access to vaccines. Similarly, if you have no bike, boat etc, you can't say green transport is within the individual. Or take diet: if healthy food is expensive, but junk food isn't, claiming a poor person is responsible for their obesity is not correct. It is never that simple as you state.
Response: authors completely agree with the referee
We modified accordingly to
“The population around the world has different behavior towards food system during COVID-19 pandemic……………………. which caused a significant weight gain”
“Thus, both individuals and governments are share in the responsibility to shift to a preventative paradigm during the pandemic or at least alleviate the impact on clinical symptoms of the case of already being infected. Governments and donors must respond quickly as soon possible to build food system [81].”
- Line 275: The most popular are communications websites such as Facebook, Instagram, Twitter, YouTube, and LinkedIn - that is quite western society centered. I would expect Alibaba to have a large if not larger community than LinkedIn. And what about TikTok?
Response: Suggestion was taken in consideration and incorporated in the manuscript
“Also, the application of TikTok has had a large share of participation and ventilation during the period of the pandemic, as many young people and even actors have resorted to it. The latest TikTok stats show that the platform has 689 million monthly active users worldwide as of January 2021[64].”
“Using social media along with artificial intelligence, the Alibaba group, have made some applications, based on cases parameters like self-reported health status, history of travels and contacts, that can identify COVID-19 cases [77]. Moreover,”
- line 283ff: Before the pandemic, R showed that most adults accessed information primarily through Facebook (79% of internet users), followed by Instagram (32%) and then Twitter (24%) [43, 63].
Here you have to distinguish information from entertainment. In a range of countries the pandemic caused many / majority to watch and consume trustworthy information sources like national broadcasting institutions. Part of those news might appear (as illegal copy in a strict sense, see Australia vs Facebook on news reporting) on social media.
Please be more balanced. Just using Facebook and Co, does not mean people trust those information or deem this relevant information (for example about taxes, food safety etc)
Response: We fully agree with the reviewer and we thank them for highlighting this issue. The reviewer comment was taken in consideration and incorporated in the manuscript as mentioned below:
“Knowledge exchanges was also needed between nations, as at least 160 countries had confirmed cases as of the beginning of 2020 (March 2020). Whether it is entertainment, reliable or fabricated information, primarily through Facebook (79% of internet users), followed by Instagram (32%) and then Twitter (24%) [43, 63]. Therefore, social media could be a suitable method to communicate the best practices for preventing COVID-19 spread. Other statistics websites and scholarly publications are known to be slow [71].”
- line 305 psychological first aid through Chabot - do you mean through chatbots? Never heard the term Chabot, please explain what this is. Thanks
Response: Changed to “chatbots”
- line 308 A leader in e-learning during the pandemic is the Zoom meetings app, which allows teachers and students to simulate a live course [70]
Are you sponsored by Zoom? Admittedly, Zoom benefited from the pandemic, but there are other tools (MS teams, CANVAS) that offer e-learning, and video-chat communication.
More crucial though, you jump from social media used for providing information to citizens to e-learning. This comes very sudden. I would expect a separate paragraph on education during the pandemic (and used technology)
Again, not all countries have the resources (read: money) to use zoom.
Response: we completely agree with the referee and we clarify the point into " 3.2. Egyptian education system as an example" section
- line 367: "graduating without studying some basic courses will affect during the actual work" - I think you mean: ... courses will have an affect on their actual work ...
Response: Adjusted
"A negative effect is expected for their scores on of the exams. For students who are about to graduate, also graduating without studying some basic courses will have an effect on their actual work each in a different field of specialization [20]."
- line 377: e-books in the form of a tablet or iPad delivered to each student - English is odd, an e-book does not have the form of a tablet or iPad - these are just means to read an e-book. Maybe say smth like: ... e-books readable on tablets (it is not needed to state iPad as an iPad is a tablet)
Response: Corrected
"One of the pioneering methods that were applied before the COVID-19 pandemic is using e-books readable on tablets e-books in the form of a tablet or iPad delivered to each student."
- line 443: The WHO Director recently announced that WHO will perform clinical trial ... this is old news, the solidarity trial started in April 2020, we now have April 2021
what you also miss is the gender imbalance. The lockdown and restrictions in academia has caused female researchers to pay a higher price (home schooling) than male researchers. The long-term consequences are fewer females in top positions, which will affect recruiting, the academic culture in general, and as you know: a country with more gender balance / equity is economically more successful (see e.g. Scandinavian countries, New Zealand etc).
Response: “Research activities and gender imbalance
The lockdown and restrictions in academia have caused female researchers to pay a higher price than male researchers. This was evident in the scientific performance of the females, which was inversely proportional to the societies' fight against COVID-19. In many countries, 50% of medical graduates are females, and 70% work within the medical sector. However, this does not appear in the publishing rates during the COVID-19 period compared to males [106]. The rate of publication as a first author for women increased from 27% to 37% in medical journals during the period from 1994 to 2014 [107]. COVID-19 threatens this progress through inflate the existing gender disparities. Gabster and col-leagues [106] conducted a statistical analysis of 159 papers on the COVID-19, and the re-sults of the total number of females participating in the research were 9% lower than in the 2018 medical journals. Consequently, it is imperative to support women in the current pandemic until it passes away and beyond. The high pressure causes women to leave academic work and look for opportunities to earn a living or work at home. This is the danger, because the scientific and medical communities need gender diversity to improve all sectors [106]."
- line 474: The acceleration in this case was due to the advent of genomics technologies and computational approaches.
This is not per se wrong, but the acceleration was mostly due to governments (EU) putting lots of money into companies developing vaccines and streamlining the bureaucracy / administration / ethical approvals that comes with vaccine development (phase I, phase II and phase III permissions).
Response: "Thus, both individuals and governments are sharing the responsibility to shift to a preventative paradigm during the pandemic or at least alleviate the impact on clinical symptoms for whom have been infected. Governments and donors must respond quickly as soon possible to build food system."
- line 482: however, as of August 11, 2020, it had not entered Phase III clinical trials - we have now April 2021, please update your manuscript.
Response: “Besides the Pfizer and BioNTech vaccine (BNT162b2), another three vaccines have been developed. The first is a DNA (human adenovirus vector) type developed by Gamaleya Re-search Institute and named Sputnik V, it has not entered Phase III clinical trials yet. …FAD [121].
- line 508: of a large drop that occurred in the atmosphere change [107–109]. either delete change or say that the change in the atmosphere was a large drop
Response: changed to "Modern technologies and satellites coupled with ground-based mapping helped researchers making fast predictions and conducting a real-time screening [115–117]."
- line 568: COVID-19 has spread worldwide since the turn of the year, taking a large number of victims, particularly in China, Italy, the U.S., Spain, Germany, and Iran [117].
This seems to be a very outdated references. Most victims are in Brazil and India. Within Europe it is neither in absolute not in relative numbers Germany (first wave it was Belgium and UK, now it is France, Poland, Estland). You can use the John Hopkins dashboard: https://coronavirus.jhu.edu/ or ourworldindata.org
Response: We agree with the reviewer and hence addressed this issue:
“6.1. During the peak period of COVID-19 outbreak
The spread of the COVID-19 pandemic began at the end of 2019 in China, and the severity of the spread around the world intensified at the beginning of 2020,”
“After the shock period and updates
After the first sudden wave, the centers of the disease spread around the world and began to change between countries. To the latest updates of April 23, 2021. The U.S. is the largest in the number of injuries, followed by India, then Brazil as a segment with the largest numbers, then France, Russia, Turkey, and the United Kingdom, respectively. Situation by region of the World Health Organization, America than Europe and south-East Asia comes the third [136]. This shift made the economies of the affected countries at the beginning start to regain equilibrium again, because of the strong lesson during the pandemic. China was the first country to recovery then throws the spillover effects countries rolled, by extension. The economic growth was 0.17% of upper-middle-income countries i.e., India, Brazil, Philippines, Indonesia, Malaysia Mexico, Peru, South Africa, Thailand, and Turkey, 0.16% in lower-middle-income countries i.e., India and the Philippines and 15% on high-income countries i.e., Australia, Canada, European Union, Japan, Korea, Norway, United Kingdom, and the USA [137]”
- line 603ff: the examples you give should be updated.
1. most of these companies received state loans
- some of these companies are back to (nearly) normal, e.g. car manufactures
entire paragraph needs updating. China is back on track, and the gross domestic product is predicted to be positive for most countries
Response: “6.2. Updates after the first wave
After the first sudden wave, the centers of the disease spread around the world and began to change between countries. To the latest updates of April 23, 2021. The U.S. is the largest in the number of injuries, followed by India, then Brazil as a segment with the largest numbers, then France, Russia, Turkey, and the United Kingdom, respectively. Situation by region of the World Health Organization, America than Europe and south-East Asia comes the third [135]. This shift made the economies of the affected countries at the beginning start to regain equilibrium again, because of the strong lesson during the pandemic. China was the first country to recovery then throws the spillover effects countries rolled, by extension. The economic growth was 0.17% of upper-middle-income countries i.e., India, Brazil, Philippines, Indonesia, Malaysia Mexico, Peru, South Africa, Thailand, and Turkey, 0.16% in lower-middle-income countries i.e., India and the Philippines and 15% on high-income countries i.e., Australia, Canada, European Union, Japan, Korea, Norway, United Kingdom, and the USA [136]."
minor typos
- line 164: [30].. - remove one "."
Response: Done
- line 169: s, , e - remove space and ,
Response: Done
- line 180 andswitching - space missing
Response: Done
- line 255 united states - United States
Response: Changed
- line 373: getaways – gateways
Response: Changed to” gateways”
- line 375: The Egyptian educational as example - presumably you want to say; the Egyptian education system as an example
Response: changed to “Egyptian education system as an example”
- line 380: on large scale with all educational stages - ... at all ...
Response: Done
- line 385: made students gains – gain
Response: Changed to “gain”
- line 593 pandemic,, th - remove "
Response: Done
- line 596: "have been effected than others" - more is missing, i.e. effected more than others
Response: Changed to “affected more than others”
- line 599: Brittan - do you mean Britain? UK?
Response: Corrected
- line 619: 2.4%t of the GDP - remove t
Response: Changed to “2.4% of the GDP”
- There are more English issues, this paper really needs proof-reading. But since you should also restructure, update and thereby rewrite many sections, I won't list all odd sentences.
Response: Authors read the manuscript thoroughly and amend it accordingly.
Reviewer 3 Report
Thank you for allowing me to review this manuscript. This study discussed the impact 39 of COVID-19 on different life aspects including; social life, health, education, environment, and economy. From this point of view is a very interesting article. However, my major concern is that the study uses a narrative approach, which may affect the results. I wonder if the authors can report how they identified the used studies. Generally, I believe that this study would be more useful based on a systematic review approach which gives the opportunity to examine the quality of the selected articles.
Author Response
Thank you for allowing me to review this manuscript. This study discussed the impact 39 of COVID-19 on different life aspects including; social life, health, education, environment, and economy. From this point of view is a very interesting article. However, my major concern is that the study uses a narrative approach, which may affect the results. I wonder if the authors can report how they identified the used studies. Generally, I believe that this study would be more useful based on a systematic review approach which gives the opportunity to examine the quality of the selected articles.
Response: We would like to thank the reviewer for the nice comment.
Our work highlighted the impact of COVID-19 on different life aspects including; social life, health, education, environment, and economy. We reported the positive and negative consequences of COVID-19 based on the published data. This work is a spontaneous continuation of pervious work published in 2020 (Khalifa et al., 2020).
Khalifa, S.A., Mohamed, B.S., Elashal, M.H., Du, M., Guo, Z., Zhao, C., Musharraf, S.G., Boskabady, M.H., El-Seedi, H.H., Efferth, T. and El-Seedi, H.R., 2020. Comprehensive overview on multiple strategies fighting COVID-19. International Journal of Environmental Research and Public Health, 17(16), p.5813.
Your suggestion will be followed in the future study.
Round 2
Reviewer 2 Report
The authors submitted a review on the pro's and con's of the pandemic. They review the consequences of a world during a pandemic by listening up the effects on lifestyle, split into
a) stay-home (social relationships and well-being),
b) transportation
c) food supply and health
d) social media (usage)
they then look at education and research, split into
a) education
b) use Egypt as an example
c) research activities
d) research activities and gender imbalances
they then look at health, split into
a) Health and psychological effects
b) Vaccine production
then they look at the weather and environment
then into the economy, split into
a) peak period of the outbreak (there are multiple)
b) after the shock
As can be seen this is an uneven structure as health / well-being comes up in three sections. Some sections are "tools" or means. A more natural (common) categorization would be (suggestion)
1. Health (includes review on changes in diet, changes in exercising, mental health including domestic violence, but also stop of vaccination programs for Malaria, Typhus, etc.)
2. Economy (includes transportation, housing market, globalization / supply chains, temporal aspect of the economy (sharp drop in employment and export but then recovery))
3. Education and Research (schools, universities, i.e. affect of home-schooling on pupils and their parents, leads naturally to gender imbalance)
4. Environment (reduced emissions but also reduced controls e.g. Amazon deforestation, less light and air pollution)
interplay between factors: increased social tension - the rich can protect themselves from the disease, the poor can't, higher death rate among African Americans, Latin-American immigrants in the US (hispanics), also in European countries immigrant groups have higher infection and death rate etc pp
Work-life style - moving out of cities etc pp
if no restructuring is done, at least there should be some logic within sections. For example 2.4 needs re-ordering. I recommend to read and apply: https://journals.plos.org/ploscompbiol/article?id=10.1371/journal.pcbi.1005619 (you don't need to cite the article, just apply those basics for your review and within sections), for 2.4.
define social media, state usage of various social media platforms, state function of social media during pandemic (pro's and con's), provide the reader with a take home message (synthesizing what you reviewed under pro's and con's)
I am not writing this section for you but in bullet point it could be (and then please apply this throughout the review, read: for all sections)
- social media = digital tools to allow social networking
- 79% use FB, 32% Instagram etc pp according to references XYZ
- both traditional news media (public broadcasters, newspapers), international organizations like the WHO, conspiracy groups, influencer, post / send information via social media.
- infodemic
- Pro's: wide reach, cheap, fast dissemination, allows social contact despite physical distancing (videochats with family etc ...
- Con's: no verification for some websites, chatbots, disinformation, ...
- my personal take home message (you can have your own): censoring should be done carefully (forbid spreading of all kinds of extremism, e.g. racism, sexism, child abise etc) rather educating citizen in social media usage (as done in the school curriculum in Finland) as the advantages (fast and easy access) are huge ...)
the open-access journal part (line 341ff) belongs better in the education and research chapter, also line 343ff on home office belongs better into economy
what is the difference between writing review articles (line 416), review knowledge in a specific filed (line 418) and conducting systematic reviews (line 419)???
line 432ff - here you talk about vaccine production, but you have this too in section 4.2. - just one very clear example of your review not being structured well.
line 511ff - given the authors background I would expect they are familiar with the vaccine of concern terminology
line 575ff - what you describe is zoonosis
line 580ff - fits better in the general discussion / conclusion as it links various categories (environment, economy, health, etc)
line 587ff - logically I would expect this far earlier if not the first chapter, as describing the temporal change lays the foundation for understanding what happens to health (humans) and the environment. As you write, the outbreak caused lockdowns and restrictions - and the lockdowns and restrictions forced people to a new lifestyle. It is NOT the other way around. So your paper should be structured accordingly
I am surprised that you only found evidence for elderly being affected by isolation / lockdown. Most literature I know of describe major mental health issues in young people (tweens).
There are major English issues, some sentences are hard to parse, incorrect use of conjunctions, past tense etc. , some examples given below
line 38, space missing between live in (it is not live-in)
line 64 served as a mean - delete "the" before "a mean", and delete "the" before states
line 73 has made changed daily life for the better or worse - should be
"changed daily life for the better or worse" i.e. delete has made
line 75 human behavior was evolved - delete was
line 85 However, ... content-wise the however is not fitting as you are not opposing the previous statement. Delete however
line 176 was included - should be were included
line 248 - recommend a line break / start a new paragraph with "The population around the world ...
line 279ff - major language issues here.
line 301 sever - should be serve
line 315 delete demonstrates
line 336 delete were
line 347 should be "staying at home during quarantine conditions" - not as you wrote: "staying at the home of quarantine conditions"
line 365 should be "at a large scale" (not of a large scale)
line 389ff repetition
line 429 in $1.8 billion in - delete the first in
line 469 "this is the danger that is imperative we avoid, ..." very odd English. Maybe you could write: An increase in gender disparity is detrimental and authorities should counteract it.
line 497 two puncutations
line 515 two "the"
line 523 "Be human ..." should be "Are human beings ..."
line 538 increased thickness - should be increased in thickness
line 614 should be "world faces a real ... " not as you wrote "world is faceda real..."
line 632ff makes the same statement as line 634
line 641ff - English not comprehensible, e.g. "Equilibrium again, ..., ... "throws the spillover effects countries rolled..." - I can guess what you want to say but please get a native English speaker to read your manuscript. Thanks
line 657 space missing after MERS.
line 673 post-pandemic (not post-panedemic)
Author Response
The authors submitted a review on the pro's and con's of the pandemic. They review the consequences of a world during a pandemic by listening up the effects on lifestyle, split into
a) stay-home (social relationships and well-being),
b) transportation
c) food supply and health
d) social media (usage)
they then look at education and research, split into
- a) education
b) use Egypt as an example
c) research activities
d) research activities and gender imbalances
they then look at health, split into
a) Health and psychological effects
b) Vaccine production
then they look at the weather and environment
then into the economy, split into
a) peak period of the outbreak (there are multiple)
b) after the shock
As can be seen this is an uneven structure as health / well-being comes up in three sections. Some sections are "tools" or means. A more natural (common) categorization would be (suggestion)
1. Health (includes review on changes in diet, changes in exercising, mental health including domestic violence, but also stop of vaccination programs for Malaria, Typhus, etc.)
2. Economy (includes transportation, housing market, globalization / supply chains, temporal aspect of the economy (sharp drop in employment and export but then recovery))
3. Education and Research (schools, universities, i.e. affect of home-schooling on pupils and their parents, leads naturally to gender imbalance)
4. Environment (reduced emissions but also reduced controls e.g. Amazon deforestation, less light and air pollution)
interplay between factors: increased social tension - the rich can protect themselves from the disease, the poor can't, higher death rate among African Americans, Latin-American immigrants in the US (hispanics), also in European countries immigrant groups have higher infection and death rate etc pp
Work-life style - moving out of cities etc pp
if no restructuring is done, at least there should be some logic within sections. For example 2.4 needs re-ordering. I recommend to read and apply: https://journals.plos.org/ploscompbiol/article?id=10.1371/journal.pcbi.1005619 (you don't need to cite the article, just apply those basics for your review and within sections), for 2.4.
define social media, state usage of various social media platforms, state function of social media during pandemic (pro's and con's), provide the reader with a take home message (synthesizing what you reviewed under pro's and con's)
Response: We would like to thank the reviewer for the nice comment.
“2.3. Transportation
Wuhan city with a population of more than 11 million ………caused by utilizing transportation and secondary pollution.”
“3.2. Social media
Social media has a wide-ranging definition, ………… but needs to be well controlled so that misinformation does not spread but rather serves as means to help.”
“5.1. Food supply, dietary patterns, nutrition, and health
In the face of the COVID-19 outbreak……. respond quickly as soon as possible to build a food system [128].”
I am not writing this section for you but in bullet point it could be (and then please apply this throughout the review, read: for all sections)
- social media = digital tools to allow social networking
- 79% use FB, 32% Instagram etc pp according to references XYZ
- both traditional news media (public broadcasters, newspapers), international organizations like the WHO, conspiracy groups, influencer, post / send information via social media.
- infodemic
- Pro's: wide reach, cheap, fast dissemination, allows social contact despite physical distancing (videochats with family etc ...
- Con's: no verification for some websites, chatbots, disinformation, ...
- my personal take home message (you can have your own): censoring should be done carefully (forbid spreading of all kinds of extremism, e.g. racism, sexism, child abise etc) rather educating citizen in social media usage (as done in the school curriculum in Finland) as the advantages (fast and easy access) are huge ...)
Response: Adjusted
- “One-fourth of Italy’s Gross Domestic Product (GDP) affected by emergency lockdown.
- Some of the largest car manufacturers in the world, suspended their production in Europe such as Volkswagen and Ferrari,
- Airbus production stopped in France and Spain.
- Five million workers in China lost their jobs.
- 85% of Norwegian Air canceled its flights and temporarily lay off 90% of staff (7,300 employees during March 16, 2020 [143].
- A result of the lockdown period of the pandemic, some famous tourist destinations as (i.e., Paris, Rome, Venice, and Madrid) are completely deserted [143].
- 5% decline of China’s industrial production during January and February [138].”
“Follow the WHO's (World Health Organization) guidelines for staying at home [4].”
the open-access journal part (line 341ff) belongs better in the education and research chapter, also line 343ff on home office belongs better into economy
Response: Adjusted
“Also, open-access journal could be freely accessed which allows for large progress to be made in regards to the scientific trials and helps some research areas to rise.”
“The modern economy is a highly complex web of interconnections between employees, firms, suppliers, consumers, banks, and financial intermediaries. If even few connections between any of these parts are interrupted by governmental disease policies, the outcome may provoke a cascading chain of disruptions [45].”
what is the difference between writing review articles (line 416), review knowledge in a specific filed (line 418) and conducting systematic reviews (line 419)???
Response: Adjusted
“Preparing and writing grants, review articles, and paper submissions [3,118,119].”
line 432ff - here you talk about vaccine production, but you have this too in section 4.2. - just one very clear example of your review not being structured well.
Response: Adjusted
“Combatting the COVID-19 pandemic is a race against time, testing the world’s ability to act quickly as the virus mutates. Subsequently, a variety of efforts have been made to create vaccines …….. are still challenges for vaccine development [134].”
line 511ff - given the authors background I would expect they are familiar with the vaccine of concern terminology
Response: Adjusted
line 575ff - what you describe is zoonosis
Response: Adjusted
line 580ff - fits better in the general discussion / conclusion as it links various categories (environment, economy, health, etc)
Response: Moved to be in “conclusion” section
“All of this prompts us to focus on what is referred to as the “Green Deal,” i.e., …………...COVID-19 is the reminder [143,144].”
line 587ff - logically I would expect this far earlier if not the first chapter, as describing the temporal change lays the foundation for understanding what happens to health (humans) and the environment. As you write, the outbreak caused lockdowns and restrictions - and the lockdowns and restrictions forced people to a new lifestyle. It is NOT the other way around. So your paper should be structured accordingly
Response: Adjusted
“2. Global economic recession “
I am surprised that you only found evidence for elderly being affected by isolation / lockdown. Most literature I know of describe major mental health issues in young people (tweens).
Response: Corrected
“Although social distance and health care strategies are the main ways to avoid the disease spread, communities have noticed that these strategies have consequences for mental health, for the child, Youngers and elders individuals, the poor, or those with limited income [141,142].”
There are major English issues, some sentences are hard to parse, incorrect use of conjunctions, past tense etc. , some examples given below
Response: The authors read the manuscript thoroughly and amend it accordingly. The manuscript was subjected for English editing.
line 38, space missing between live in (it is not live-in)
Response: Changed
line 64 served as a mean - delete "the" before "a mean", and delete "the" before states
Response: Changed
line 73 has made changed daily life for the better or worse - should be
"changed daily life for the better or worse" i.e. delete has made
line 75 human behavior was evolved - delete was
Response: Adjusted
line 85 However, ... content-wise the however is not fitting as you are not opposing the previous statement. Delete however
Response: Deleted
line 176 was included - should be were included
Response: Changed
line 248 - recommend a line break / start a new paragraph with "The population around the world ...
Response: Adjusted
line 279ff - major language issues here.
Response: Adjusted
line 301 sever - should be serve
Response: Changed
line 315 delete demonstrates
Response: Adjusted
line 336 delete were
Response: Adjusted
line 347 should be "staying at home during quarantine conditions" - not as you wrote: "staying at the home of quarantine conditions".
Response: Changed
line 365 should be "at a large scale" (not of a large scale)
Response: Changed
line 389ff repetition
Response: Adjusted
line 429 in $1.8 billion in - delete the first in
Response: Deleted
line 469 "this is the danger that is imperative we avoid, ..." very odd English. Maybe you could write: An increase in gender disparity is detrimental and authorities should counteract it.
Response: Changed
line 497 two puncutations
Response: Adjusted
line 515 two "the"
Response: Adjusted
line 523 "Be human ..." should be "Are human beings ..."
Response: Adjusted
line 538 increased thickness - should be increased in thickness
Response: Changed
line 614 should be "world faces a real ... " not as you wrote "world is faceda real..."
Response: Adjusted
line 632ff makes the same statement as line 634
Response: Deleted
line 641ff - English not comprehensible, e.g. "Equilibrium again, ..., ... "throws the spillover effects countries rolled..." - I can guess what you want to say but please get a native English speaker to read your manuscript. Thanks
Response: Adjusted
line 657 space missing after MERS.
Response: Adjusted
line 673 post-pandemic (not post-panedemic)
Response: Adjusted
Reviewer 3 Report
Thank you for the revised version. Despite I strongly believe that a systematic review could provide a piece of more valuable information, the author made several developments in the manuscript.
Author Response
Comments and Suggestions for Authors
Thank you for the revised version. Despite I strongly believe that a systematic review could provide a piece of more valuable information, the author made several developments in the manuscript.
Response: The authors agree with the referee and we will consider your suggestion for the coming review.